# Dynein-Inspired Multilane Exclusion Process with Open Boundary Conditions

**DOI:** 10.3390/e23101343

**Published:** 2021-10-14

**Authors:** Riya Nandi, Uwe C. Täuber

**Affiliations:** 1Department of Genetics and Evolution, University of Geneva, 1205 Geneva, Switzerland; riya.nandi@unige.ch; 2Department of Physics (MC 0435) & Center for Soft Matter and Biological Physics, Faculty of Health Sciences, Virginia Tech, Blacksburg, VA 24061, USA; tauber@vt.edu; 3Department of Bioengineering, University of Illinois at Urbana-Champaign, Urbana, IL 61801, USA

**Keywords:** dynein motors, exclusion process, phase diagram, dwell time distribution

## Abstract

Motivated by the sidewise motions of dynein motors shown in experiments, we use a variant of the exclusion process to model the multistep dynamics of dyneins on a cylinder with open ends. Due to the varied step sizes of the particles in a quasi-two-dimensional topology, we observe the emergence of a novel phase diagram depending on the various load conditions. Under high-load conditions, our numerical findings yield results similar to the TASEP model with the presence of all three standard TASEP phases, namely the low-density (LD), high-density (HD), and maximal-current (MC) phases. However, for medium- to low-load conditions, for all chosen influx and outflux rates, we only observe the LD and HD phases, and the maximal-current phase disappears. Further, we also measure the dynamics for a single dynein particle which is logarithmically slower than a TASEP particle with a shorter waiting time. Our results also confirm experimental observations of the dwell time distribution: The dwell time distribution for dyneins is exponential in less crowded conditions, whereas a double exponential emerges under overcrowded conditions.

## 1. Introduction

Molecular motors are enzymatic protein molecules that use chemical energy released from the hydrolysis of an adenosine triphosphate (ATP) molecule to drive cellular transport along cytoskeletal filaments [1]. Among the wide variety of molecular motors, linear ATP-driven motors include myosins, kinesins, and dyneins that are responsible for various cellular functions, including intracellular vesicle transport [1,2]. kinesins and dyneins move along a microtubule, whereas myosins move on an actin filament. The kinetic and mechanistic processes involved in the movement of the motors are inherently stochastic, which arises due to various interactions among the motors themselves as well as environmental factors such as track length, amount of cellular cargo acting as load, available ATP concentrations, etc. [3]. The stochasticity present at the cellular level gives rise to experimental complexity at various levels. Consequently, incorporating stochastic features into theoretical modeling of molecular motor transport becomes essential to gain adequate statistical understanding of the ensuing fluctuations and their effects. Specifically, modeling such biological transport phenomena with the help of driven diffusive systems has been fruitful in capturing many essential properties of real systems [4,5,6,7]. Various versions of the (totally) asymmetric exclusion process or (T)ASEP have been developed involving random walkers with hard-core interactions to mimic the driven non-equilibrium dynamics of kinesin and myosin [4]. In contrast, theoretical studies of dynein motors remain mostly unexplored to date, whence this constitutes the present work’s central theme.

The most exciting feature that differentiates dynein motors from kinesin or myosin is their ability to vary their step size from 8 nm to 32 nm [8,9]. In contrast with the kinesin and myosin families of motor proteins, dyneins are structurally much more complicated. Detailed experimental studies providing a dynamical understanding of dynein motion are still lacking. However, some recent experiments have been able to capture the dynamics of dynein motors over a microtubule. Interestingly, kinesins and dyneins show small but discernible sidewise fluctuations on the microtubule tracks consisting of multiple protofilaments [10,11,12,13,14]. Thus, modeling of kinesin dynamics with the ability to switch lanes has been considered [15], whereas the emergent behavior due to incorporating multi-lane dynamics of dynein motors along with variable steps sizes is as yet unknown. Most of the theoretical modeling of dynein motors has been restricted to the analysis of single-molecule dynamics [16,17], or of multi-particle dynamics on a single protofilament track [18,19].

In the present work, we investigate the combined effects of a dynein’s internal state-dependent hopping as well as hindrance caused by the presence of other motors in its path, accounting for the possibility of bypassing traffic via separate lanes. We explore the emergent features of the collective dynein dynamics by modeling the motor proteins as hard-core particles in a quasi-two-dimensional topology. This work thus considerably expands on our earlier studied model [19] of dynein motors on a one-dimensional lattice endowed with varied stepping behavior, to now explicitly include off-axis hops. We quantify the effects of the off-axis movement by exploring the phase diagram as function of the influx (α) and outflux (β) rates of the molecular motors into and out of the lattice that describes the microtubule lanes. In the presence of low- and intermediate-load environments where a motor is more likely to take longer jumps (in experiment corresponding to steps of 32 nm), we observe a first-order transition separating the high- and low-density phases for all values of the flux imbalance parameter α−β. This discontinuous phase transition can be readily attributed to the two-dimensional geometry. Environmental factors such as ATP availability and the crowding environment add stochasticity to the dynein stepping behavior, and its effect on dynein dynamics has been characterized through dwell-time distributions in recent experiments [10,11,12]. Our simulation results further confirm these experimental observations for the dynein motor dwell-time distribution, which takes an exponential functional form in less-crowded situations, whereas it turns into a double exponential under overcrowded conditions.

## 2. Model Description and Dynamics

Our model is defined on a cylindrical lattice with open boundaries at the longitudinal ends. We refer to the cylinder is quasi-two-dimensional, as we keep the longitudinal length $L_{x}$ of the lattice much larger than its transverse length $L_{y}$, $L_{y}/L_{x}\ll 1$. The quasi-two-dimensional cylinder is structurally similar to the open tube-like configuration of a microtubule, where each row or lane mimics a protofilament of the microtubule. Each lattice site on the cylinder can be occupied by at most one indistinguishable hardcore particle. A motor particle can only enter from the first lattice site of any lane with a constant rate α, and exit from the last longitudinal site of any lane with the rate β. In the bulk, a particle may move in one of the three following directions: forward along the length of the lattice, upward or downward in the transverse directions. Guided by recent experimental results that demonstrate the transverse off-axis motion of the dynein motors to be significantly slower than their forward movement [10,11,12], we choose the probability of moving in the transverse direction $P_{y}$ to be significantly smaller than the chance of forward movement $P_{x}$: $P_{x}=1-2P_{y}=0.95$, $P_{y}=0.025$. Further, depending on the load (vacuoles or ATPs) attached to the motor particles, they can advance between one and up to four lattice sites along any of the three directions, provided that the destination sites are empty. If during its hop attempt, a motor happens to reach the last longitudinal site (at $L_{x}$) prior to completing its assigned jump, it will directly exit from the lattice with a rate β.

We are interested in studying the collective kinetics of dynein motors, well-known for their dynamical jump length, which ranges from 8 to 32 nm depending on various factors such as the available ATP concentration, amount of load, etc. [8,9]. We ignore the structural complexity of the dynein molecule and consider the dimeric dynein motor as a hardcore particle with four ATP binding sites. One primary ATP binding subsite is responsible for hydrolysis, while the remaining three secondary binding subsites carry the load. The following three distinct internal processes determine the probability of performing a number $n_{s}$ of steps:(i)**Attachment:** One unit of ATP attaches to the empty primary binding site with a fixed probability $P_{att}$ or to any of the available secondary sites with an also constant probability, $S_{att}$.(ii)**Detachment:** One unit of ATP detaches either from the primary site with the fixed rate $P_{det}$ or from one of the three secondary sites with the constant probability $S_{det}$.(iii)**ATP hydrolysis:** An occupied primary site hydrolyses ATP to ADP to generate mechanical energy that enables it to perform movement. This ATP binding induces dissociation of the motor from the microtubule. After detaching from the microtubule, the motor rearranges its structure and becomes poised for the powerstroke movement, triggered by the ATP hydrolysis. This structural change is followed by a diffusional search for the target binding site over the microtubule [20,21]. Thus, if the primary binding site is occupied, the motor attempts to hop (4−s) steps, where *s* is the number of secondary sites holding ATP (thus, *s* can be either 1, 2, or 3).

To incorporate excluded-volume interactions, we consider the sliding behavior of a motor particle to an empty site which prohibits the particles from overtaking in the same lane. However, they can bypass other particles present in different lanes. A schematic of the model is illustrated in Figure 1. The rate of hopping different step sizes depends on the attachment and detachment probabilities. Specifically, the rate $R_{i}$ to jump *i* steps (given there are no obstructions due to the presence of other particles) is given by
(1)R4=Patt(1−Pdet)∑i(SattSdet)i(1−Satt)3−i,(2)R3(3)R2=Patt(1−Pdet/4)Satt2(1−Sdet)2[(SattSdet)+(1−Satt)],(4)R1=Patt(1−Pdet)Satt3(1−Sdet)3.
Further, due to excluded-volume constraints, the rates get modified by the presence of other particles on the path. The stationary state current across the bonds i−1 and *i* is a sum of all possible jumps, given by
(5)$$\begin{array}{cc} j(\rho ) & =\langle \left(R_{1}+R_{2}+R_{3}+R_{4}\right)\eta_{i-1}\left(1-\eta_{i}\right)\eta_{i+1}+\left(R_{2}+R_{3}+R_{4}\right)\eta_{i-1}\left(1-\eta_{i}\right)\left(1-\eta_{i+1}\right)\eta_{i+2}\\ \; & +\ldots +R_{2}\eta_{i-1}\left(1-\eta_{i}\right)\left(1-\eta_{i+1}\right)\sum_{l=0}^{\infty }\eta_{i+l+2}\prod_{k=1}^{l}\left(1-\eta_{i+1+k}\right)+\ldots \rangle \;, \end{array}$$
where $\eta_{j}$ denotes the occupancy of site *j*. Within a standard mean-field approximation, one obtains the probabilities to move provided the empty space in front of the particle is greater than (i−1) as
(6)$$v_{4}=R_{4},\;v_{3}=R_{4}\rho +R_{3},\;v_{2}=\left(R_{4}+R_{3}\right)\rho +R_{2},\;v_{1}=\left(R_{4}+R_{3}+R_{2}\right)\rho +R_{1}.$$
Here, ρ denotes the mean particle density in the bulk and the mean-field bulk current is approximated as (7)$$J\approx \sum_{i=1}^{4}i\;v_{i}\;\rho {\left(1-\rho \right)}^{i}.$$
Before starting the discussion of the results for the dynamics of this dynein transport model, we provide a brief review of the well-known stationary-state results of the one-dimensional TASEP model with open boundaries [22] in the following subsection.

### Phase Diagram of the One-Dimensional TASEP with Hop Rate r

In the case of a one-dimensional TASEP, where a hardcore particle can move to its nearest neighboring site with a constant hop rate *r*, the steady-state particle current for periodic boundary conditions is given as J=rρ(1−ρ) at all lattice sites. However, for open boundary conditions, depending on the influx (α) and outflux (β) rates from the first and last sites of the lattice, the TASEP stationary states can be classified into three different phases: low-density (LD), high-density (HD), and maximal-current (MC). From the asymptotic analysis performed in earlier studies (please see, e.g., Ref. [22] for details), the characteristic features of the three phases are as follows [22,23]:(a)**Low-density phase (LD)**: In this phase, the bulk stationary density saturates to a constant value $\rho_{LD}=\alpha /r$, which yields the stationary current $J_{LD}=\alpha \left(1-\rho_{LD}\right)=\alpha \left(1-\alpha /r\right)$. Near the exit boundary, the density corresponds to $\rho_{L_{x}}=J_{LD}/\beta$.(b)**High-density phase (HD)**: Here, the bulk stationary density is $\rho_{HD}=1-\beta$, thus the steady-state particle current is given by $J_{HD}=r\beta \left(1-\beta \right)$.(c)**Maximal-current phase (MC)**: The maximal particle current is obtained when the density in the bulk is 1/2 and hence $J=J_{max}=r/4$ for α>r/2 and β>1/2. Near the boundaries, the density deviation from its bulk value shows power law decay, which indicates the presence of long-range correlations in the system.

The transition across the LD and HD phases is first-order (discontinuous); hence, a coexistence phase is observed along the phase transition line. Equating the LD and HD particle fluxes, one finds the coexistence line located at β=α/r, as is confirmed by numerical simulations. Furthermore, these results hold even for the two-dimensional TASEP with periodic boundary conditions in the transverse direction.

## 3. Results

Compared to the one-dimensional TASEP, the dynamics of the dynein particles in both the one-dimensional and quasi-two-dimensional settings are considerably more complicated, hence it is non-trivial to extract any precise analytical understanding. Consequently, we employ extensive numerical simulations to explore the dynamics under various contrived environmental conditions. The results shown in the following sections pertain to lattices with dimensions $L_{x}=1000$ in the one-dimensional case, and $L_{x}=1000$, $L_{y}=4$ (i.e., four adjacent microtubule lanes) for the quasi-two-dimensional topology. The ATP attachment and detachment rates for the primary site are chosen as $P_{att}=0.8$ and $P_{det}=0.2$, respectively. We worked with three different load conditions for each motor as determined by the load attachment rate $S_{att}$ and detachment rate $S_{det}=1-S_{att}$ at the secondary sites. We selected the following set of binding rates for each secondary site:$$S_{att}=\left\{\begin{array}{cc} 0.8, & \mathrm{high}\text{-}\mathrm{load}\;\mathrm{conditions},\\ 0.5, & \mathrm{intermediate}\text{-}\mathrm{load}\;\mathrm{conditions},\\ 0.2, & \mathrm{low}\text{-}\mathrm{load}\;\mathrm{conditions}. \end{array}\right.$$

### 3.1. Phase Diagram for Dynein Particles in One Dimension

We aim to evaluate the stationary state phase diagram for the dynein particle exclusion process in one dimension. To that end, we first write down the mean-field bulk current that has been formulated in our previous work, with periodic boundary conditions [19]:(8)$$J=\rho \sum_{i=1}^{4}i\;v_{i}\;{\left(1-\rho \right)}^{i},$$
where $v_{i}$ denotes the probability of taking *i* steps, and explicitly depends on the attachment and detachment of ATPs to their primary and secondary binding sites, as shown in Equation (Equation 6). Then the bulk density $\rho_{max}$ in the MC phase is obtained from the maximum of the mean-field current, $\frac{\partial J}{\partial \rho }{|}_{\rho_{max}}=0$, where *J* is given by Equation (Equation 8), for different load conditions. Straightforward algebraic calculations further determine the exact values (within this mean-field approximation) for the rates α and β beyond which the stationary-state dynamics resides in the maximal-current (MC) phase. The resulting transition point for the MC phase is (9)$$\begin{array}{c} \alpha >2\rho_{max}\sum_{i=1}^{4}i\;v_{i}\;{\left(1-\rho_{max}\right)}^{i-1},\;\;\;\;\beta >1-\rho_{max}. \end{array}$$

This analysis of the MC transition point is in excellent agreement with our simulation results, as marked by the dashed line in Figure 2, both for high- and low-load conditions. Our findings from the Monte Carlo simulations and data analysis demonstrate that for the one-dimensional dynein particle dynamics, one observes a nonlinear first-order transition line (coexistence line) between the LD and HD phases, in contrast with the linear coexistence line for the one-dimensional simple TASEP, which is plotted as the (red) dotted line in Figure 2 for comparison. The bulk density in the LD phase is proportional to the injection rate α, and shows intriguing periodic oscillations at the boundaries [19], which however emerge as artifacts of the long jumps. In much larger systems, we do not observe the propagation of these oscillations from the exit boundary. In the HD phase, the bulk density is precisely 1−β, as in the one-dimensional TASEP. The density profiles in all three distinct nonequilibrium phases are shown in the insets of Figure 2. These insets also display the jump length statistics in the different phases for high- and low-load conditions. It is apparent in Figure 2a that the statistics of taking single steps under high-load conditions is significantly high, so that the results for the stationary-state bulk density and current become similar to the TASEP. It is important to note that the increased probability for long jumps reduces the phase space area of the maximal-current phase. Long jumps break particle-hole symmetry and promote the motor particles’ nonuniform distribution over the lattice.

### 3.2. Phase Diagram for Dynein Particles in Two Dimensions

This section discusses our simulation results for the quasi-two-dimensional setting on a rectangular lattice of size $10^{3}\times 4$ and compares them with the one-dimensional case (where also $L_{x}=10^{3}$). We start with analyzing the jump statistics and the dynein particle current along the longitudinal direction. The stationary-state current in the transverse direction always vanishes for an infinitely large system owing to the periodic boundary conditions and our choice of equal hopping probabilities for up- and down-moves.

Under high-load conditions, the jump size statistics show that the number of nearest-neighbor hops is more significant than the number of longer-distance jumps, as seen in Figure 3d. One would thus expect the ensuing dynamical steady-state phase diagram to resemble that of the one-dimensional TASEP. Indeed, akin to the TASEP, we observe the presence of all three phases LD, HD, and MC in the α−β parameter space as shown in the inset of Figure 4a, but with a shift in the transition line. Owing to the small but nonzero number of long jumps performed by the dynein particles, the MC critical point becomes shifted toward a higher value of the influx rate α. In the inset of Figure 4a, the solid gray line represents the TASEP transition lines, while the broken black lines indicate the phase boundaries obtained for our quasi-two-dimensional dynein model under high loads. In addition, we recall that the MC phase features long-range correlations. Hence, the density profile in the MC phase is in fact independent of the rates α,β and displays a tangent profile $\rho \left(x\right)\approx \rho_{max}\left(1-qtan\left[q\left(x-x_{0}\right)\right]\right)$, where $q=\sqrt{J(L)/J-1}$, with power law behavior near both boundaries, yet with a decay exponent 1/2 unlike the linear decay familiar from the standard TASEP. In the coexistence phase, where the particle density is in the mixed state, we observe a hyperbolic tangent density profile. Moreover, the density profile amplitude in the LD phase is proportional to the rate α and shows damped oscillatory behavior close to the entrance boundary. These spatial oscillations have a periodicity of four sites, a qualitatively quite similar feature as for the one-dimensional dynein particle model discussed in our previous work [19]. Further, the bulk density in the HD phase is always approximately near (1−β) as demonstrated in Figure 3a. Interestingly, under both low- and intermediate-load conditions, we only observe two stationary-state phases with a first-order transition line separating them. The maximal-current phase is absent, as shown in the main Figure 4a. Here, it is worth mentioning that our numerical data suggest the recovery of the MC phase, once the chosen probability to move in the transverse direction $P_{y}$ is set to be of the same order as for longitudinal jumps $P_{x}=\left(1-2P_{y}\right)$, as depicted in Figure 4b for low-load conditions. Under these conditions, the jump length histogram implies that the probability of taking longer jumps is more significant in the LD phase, whereas shorter jumps dominate in the HD phase. However, the magnitude of this difference is not as substantial as for high loads, as can be seen in Figure 3e,f. The farther jumps and the small side-wise fluctuations in the transverse direction in this two-dimensional setting allow a dynein motor to bypass other particles via different lanes. Hence the exclusion interactions do not facilitate the system to build long-range particle correlations. A similar absence of the maximal-current phase has also been observed in a TASEP model with hierarchical long-range connections defined on a network [24].

The density in the LD phase is again proportional to the influx rate α, and the bulk density in the HD phase is approximately close to (1−β), as demonstrated in Figure 3b,c. The results depicted in Figure 5 support the discontinuous (first-order) character of the transition between the LD and HD phases for the entire parameter space. We observe a continuous change in the stationary motor particle current with the influx and outflux rates as evidenced in the main Figure 5, which however causes an abrupt change in the density profile, noticeable in the inset of Figure 5). Before exploring the collective dynamical behavior of the dynein particles in the quasi-two-dimensional lattice geometry, we would like to emphasize that our simulations indicate that an increase in the number of lanes does in fact not lead to any qualitative change in the results.

### 3.3. Dynamics of Dynein Particles in Two Dimensions

To understand the change in the dynamics caused by the quasi-two-dimensional geometry, we first explore the single-run statistics under various load conditions in the distinct phases. Specifically, in Figure 6 we show the dynamics of three different tagged dynein motor particles: A particle is tagged at different entry times $t_{en}$, which effectively corresponds to different crowding environments until the system reaches its stationary state. “Particle A” denotes the first particle that enters the lattice. “Particle B” and “Particle C” enter the system at times proportional to $L_{x}$ and $L_{x}^{2}$. Examples for single-run trajectories of particles tagged in this manner are displayed in Figure 6. Irrespective of α and β, particle A is more likely to move forward while remaining in the same lane than particles B and C. It is less probable for particle A to encounter obstacles (other motor particles), and hence sideways fluctuations are less prominent in its trajectory than for the later-arriving particles. However, as the overall particle density increases with time, the kinetics becomes increasingly obstructed, and individual particles are forced to switch to other lanes, as is apparent in the more helical structures visible in the trajectories of particles B and C in Figure 6. We also observe that the later particles take longer to exit from the last lattice sites due to crowding. Following these observations, we analyze the trajectories’ mean-square displacement (MSD) to better understand the slowing-down of the motor particles transport, and to provide additional information on the changing time scales under different conditions.

We have thus numerically evaluated the mean-square displacement (MSD) of a tagged particle. A particle is tagged upon its time of entry; this corresponds to random initial conditions in different runs. The MSD in the transverse direction is always diffusive, owing to the symmetry in the probabilities of up- and downward movement. However, the dynamics along the longitudinal direction clearly shows modifications in the relevant characteristic time scales. The dynamical rules assigned to a particle do not depend on the lane in which it resides. Hence, it is inherent to have similar behavior in each lane, as can also be inferred from the space-time profile of each lane shown in Figure 7a (high-density phase) and Figure 7b (low-density phase) under low-load conditions. Hence we present the MSD results in longitudinal (drive) direction which are averaged over all the lanes. It is well-known that wide variations of the TASEP exhibit dynamics that follow the scaling exponents of the Burgers/KPZ universality class in one dimension, and correspondingly the MSD grows with time as $t^{2/3}$ [25,26]. In the two-dimensional TASEP, the MSD growth with time becomes linear with logarithmic corrections [27]. For the dynein particles in our model, the simulation data yield that the long-time behavior of the MSD obeys a power-law growth with logarithmic corrections in the form $t^{\xi }/\ln t$, where ξ takes values in the range between 3/2 to 1 depending on the crowding conditions over the lattice. Irrespective of the load conditions, the dynamics for a single dynein particle is superdiffusive, and the MSD along the drive direction grows algebraically with time with an exponent 3/2 as shown in the insets of Figure 8. However, in a highly crowded environment, the particle kinetics slows down, and ξ approximately takes a value closer to unity, as evident in the main panels of Figure 8. The MSD dynamics turns out to be essentially independent of the details of the boundary injection and exit rates.

The change in the MSD growth power laws is solely due to the difference in the passage times that particles take to move from one site to another in the presence of other particles. Thus, measuring the particle dwell times adds further information about this mutually constrained particle dynamics. The dwell-time distribution is in fact an experimentally relevant quantity and has been successfully measured in a number of experiments [10,28]. In our simulations, we define the dwell or waiting time as the temporal duration, in units of Monte Carlo steps (MCS), that a particle waits between two consecutive jumps. We have numerically obtained the dwell-time statistics under different crowding conditions, as plotted in Figure 9. For particle A, which invariably encounters a less-crowded environment, the waiting-time distribution for a tagged particle decays exponentially, irrespective of the influx and outflux rates α and β, as seen in Figure 9. However, with increasing particle density across the lattice, we observe longer waiting times and slower decay of the associated dwell-time distribution with longer tails of the exponential function. The dwell-time distribution in the overcrowded environment displays a “double-exponential” form with an initial slower decay rate for shorter waiting times, which subsequently crosses over to a faster decay for longer waiting times, as seen in the main Figure 9, which corresponds to an average particle density ≈0.85. In addition, we observe intriguing oscillatory features for particle-type B (with $t_{en}\propto L_{x}$). These oscillations are prominently visible only if the average density is greater than half-filled, and represent a dynamical phase with many alternating high-density peaks and low-density valleys, akin to coexistence phases.

## 4. Conclusions

In this paper, we describe a variation of the asymmetric exclusion process with short- and long-distance jumps to model the dynamics of dynein motors, and explored the ensuing collective dynamics in one dimension as well as in a quasi-two-dimensional topology with open boundary conditions. Our work highlights the non-trivial effects of geometry in out-of-equilibrium driven diffusive systems that may modify the phase diagram and change the nature of the accompanying phase transitions. Significant changes observed in the phase diagram can be attributed to the breakdown of particle-hole symmetry for the dynein particle dynamics. In one dimension, we found a reduced maximal-current (MC) phase region in α−β parameters space due to the presence of long jumps. Using the mean-field expression for the resulting current, we present a precise lower bound for the influx and outflux rates for the maximal-current phase. We extended our study to the quasi-two-dimensional case with negligible small transverse hop probabilities. In this setting, the phase diagram shows drastic changes; we observe the disappearance of the maximal-current phase for low- to medium-load conditions caused by the mutual bypassing of the motor particles via different lanes. Moreover, these results remain qualitatively unchanged upon increasing the number of lanes beyond two. We have also presented some preliminary results for the dependence of the phase diagram on the respective probabilities of selecting different hopping directions. Interestingly, the system reverts to the phase diagram that includes a maximal-current phase when the probabilities to move along the longitudinal and transverse directions are of the same order. These approximately equal probabilities to move in either direction effectively suppresses the particle’s chances to bypass other particles blocking its large forward motions, which induces the re-emergence of the maximal-current phase. We plan to explore this feature more in the future. We hope these intriguing results will motivate further studies into the nontrivial effects of higher dimensions in variants of the TASEP model and serve as a reference for a wider range of transport processes that involve particles bypassing each other, such as vehicular traffic models and models for certain regulatory genes.

Further, we have also examined the dynamics of the dynein particles under different environmental conditions and measured their MSD and dwell-time distributions. Due to the two-dimensional topology, a dynein particle shows logarithmic corrections in the superdiffusive time-dependent growth of the MSD. Our study also demonstrates the slowing-down of the kinetics due to crowded conditions, leading to dynamics closer to diffusive behavior. Additionally, we observe consistent features in the associated dwell-time distributions. Dwell-time distributions are directly experimentally measurable and can be easily verified for different control parameters. We confirmed the effects of crowding and variable-stepping behavior of dyneins on their dwell-time distributions: We found that under less-crowded conditions, the dwell times are distributed exponentially; whereas in a more crowded environment, the waiting time distribution changes to a double exponential with two characteristic time scales. Both exponential and double exponential distributions of the dwell times have been observed in real experiments under conditions when the dynein takes longer and shorter jumps, respectively, [10,28], c.f. specifically Figure 5 in Ref. [10]. 

## Figures and Tables

**Figure 1 entropy-23-01343-f001:**
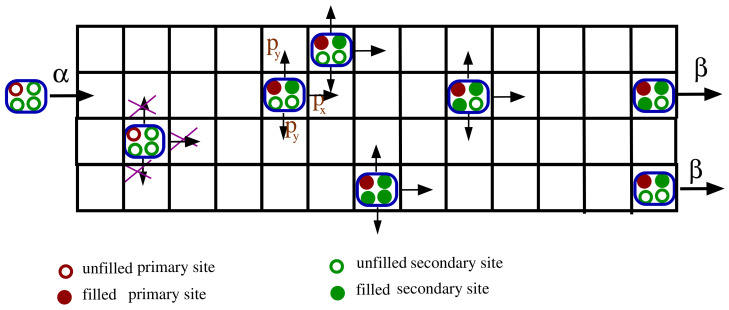
A schematic representation of the dynamics of dynein motors over a tube. We have an open longitudinal boundary through which a motor particle enters from the left edge with the rate α, and exits on the right with the rate β. Periodic boundary conditions are employed along the transverse direction. Each particle comprises four internal sections representing the primary (red circle) and secondary (green circle) ATP binding sites. A motor particle may move in any of the three forward and transverse directions, provided the primary ATP binding site is filled. The probability of hopping in the up or down direction is the same, $P_{y}$, whereas the likelihood of forward motion is set to $\left(1-2P_{y}\right)$.

**Figure 2 entropy-23-01343-f002:**
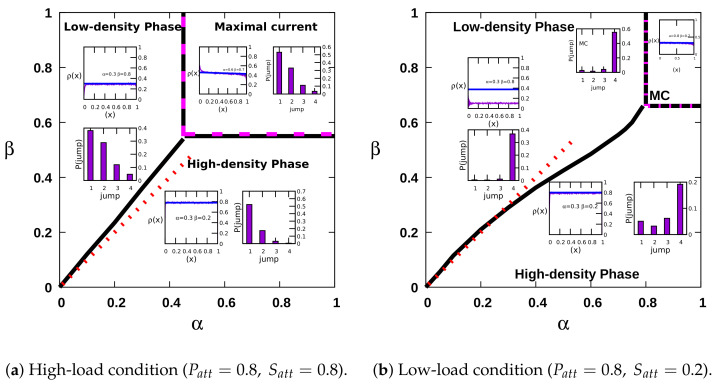
Phase diagram in the influx–outflux rate space for the dynamics of dynein particles in one dimension. The phase diagram under high-load conditions closely approximates the standard TASEP phase diagram as indicated by the red dotted line. However, the maximal current region is reduced to a small portion of the phase space under the low-load condition. The insets display the approximately TASEP-like density profile. The jump length statistics in the respective phases are also shown in the insets. In the MC phase, a reduction in the stationary density is observed upon increasing the probability of long jumps; thus, $\rho_{MC}\approx 0.45$ for high-load and $\rho_{MC}\approx 0.34$ for low-load conditions as indicated by the solid green lines in the insets of Figure 2a,b, respectively. The (magenta) broken line indicates the calculated HD/MC and LD/MC transition boundaries obtained from Equation (Equation 9) in the maximal-current phase.

**Figure 3 entropy-23-01343-f003:**
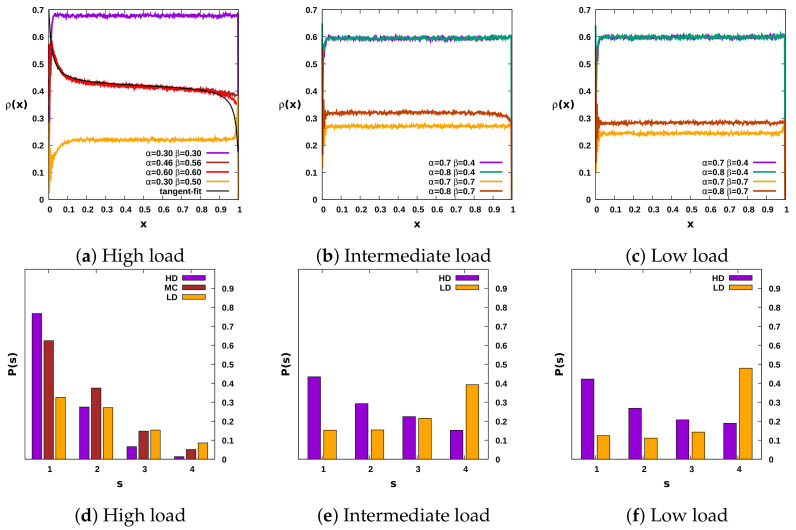
Two-dimensional lattice: The upper panels display the density profile in different phases for the three distinct load conditions. The lower panels provide the statistics of the number of steps taken by the dynein particles. Under high-load conditions, the count of smaller steps is significantly larger (Figure 3d), and the solid black line represents the best fit to a tangent density profile, $\rho_{max}(1-a\;q\tan\left[a\left(x-0.5\right)\right]$, for the MC phase, similar to the TASEP, as shown in Figure 3a. For both medium- and low-load conditions, the bulk density in the high-density phase is (1−β), whereas it is proportional to the injection rate α in the low-density phase. The number of lanes set for these graphs is four.

**Figure 4 entropy-23-01343-f004:**
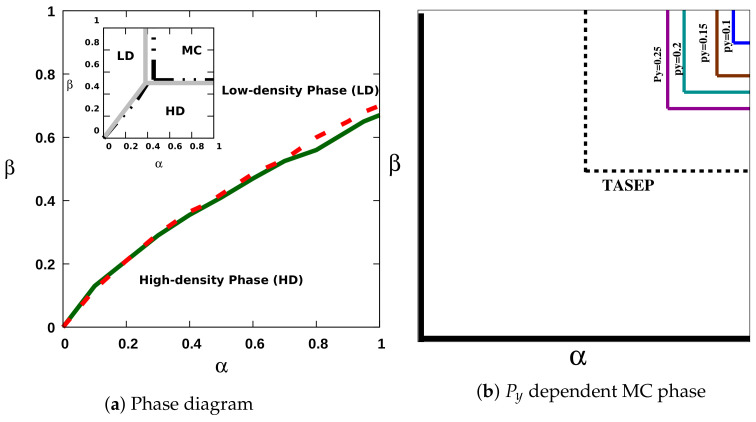
The graphs depict the steady-state properties obtained from the dynamics of the dynein particles in the quasi-two-dimensional setting. The main Figure 4a shows the phase diagram with a first-order transition separating the low- and high-density phases under low- (red dashed line) and intermediate-load (green line) conditions for small transverse jump rate $P_{y}=0.025$. Figure 4b shows the schematic dependence of the phase diagram on the probability to select the hopping direction $P_{att}=0.8,\;S_{att}=0.2$ for low loads. The values of α and β beyond which we observe the maximal-current phase for various $P_{y}$ are $P_{y}=0.1:\;\alpha >0.95,\;\beta >0.9$ (blue line), $P_{y}=0.15:\;\alpha >0.9,\;\beta >0.8$ (brown), $P_{y}=0.2:\;\alpha >0.8,\;\beta >0.75$ (green), $P_{y}=0.15:\;\alpha >0.75,\;\beta >0.7$ (purple). The dotted (black) line indicates the transition between the MC/LD and MC/HD phases for the one-dimensional TASEP.

**Figure 5 entropy-23-01343-f005:**
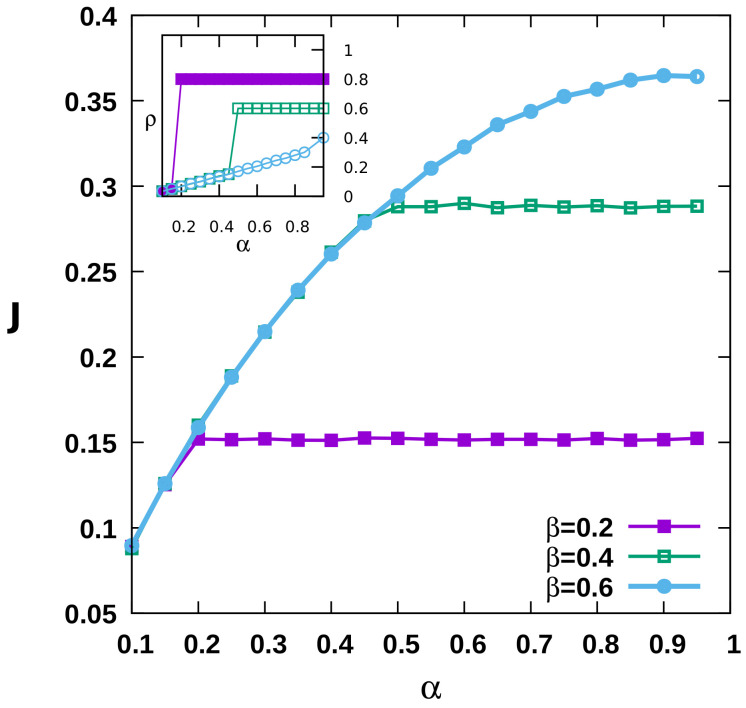
The main part of the plot shows the current profiles as a function of the influx rate α, for three different outflux rates β=0.2,0.4,0.6, under low-load conditions; the inset depicts the corresponding density profiles illustrating the discontinuous transition from the low- to the high-density state.

**Figure 6 entropy-23-01343-f006:**
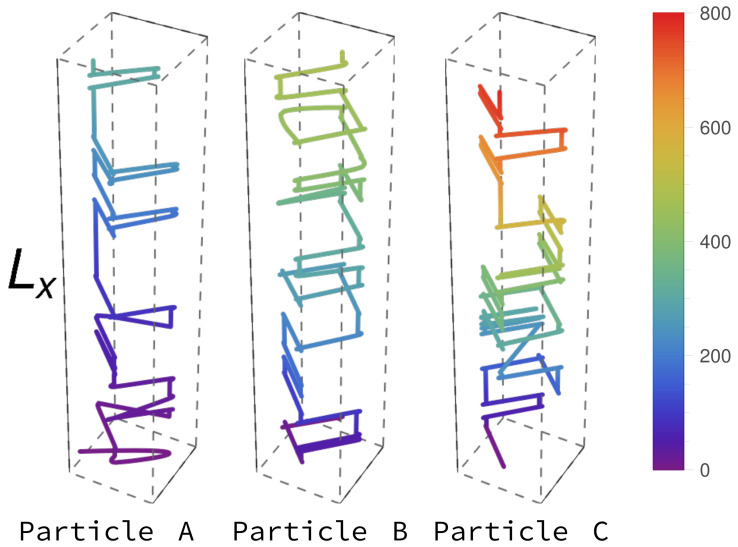
Single tagged dynein particles’ trajectories under high-load conditions. Particles A, B and C enter at t=0, $t\approx L_{x}$ and $t\approx L_{x}^{2}$, respectively. The influx and outflux rates used here are α=0.7,β=0.7, corresponding to the low-density phase.

**Figure 7 entropy-23-01343-f007:**
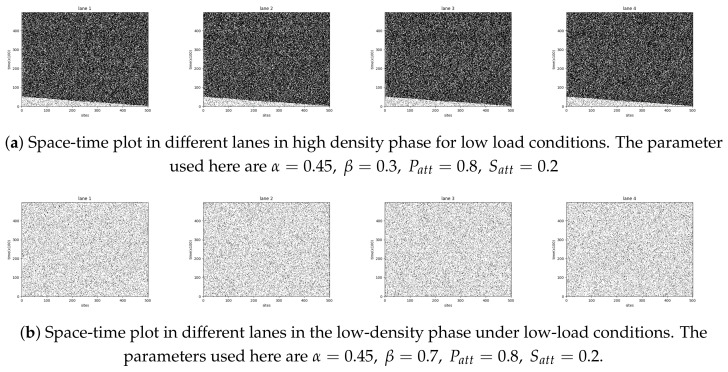
Space-time plot in different lanes under low-load conditions.

**Figure 8 entropy-23-01343-f008:**
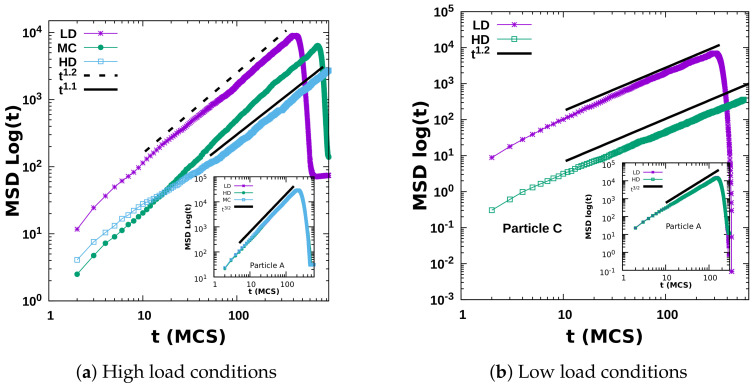
The time-dependent growth of the mean-square displacement (MSD) for the tagged particles’ C (in the main figure panels) and A (in the insets) for different phases on a finite quasi-two-dimensional lattice. The different time regimes are also marked; here, L=500, $P_{att}=0.8$, and $S_{att}=0.8$ representing a high-load, and $S_{att}=0.2$ for a low-load condition. The results were averaged over $10^{3}$ different realizations.

**Figure 9 entropy-23-01343-f009:**
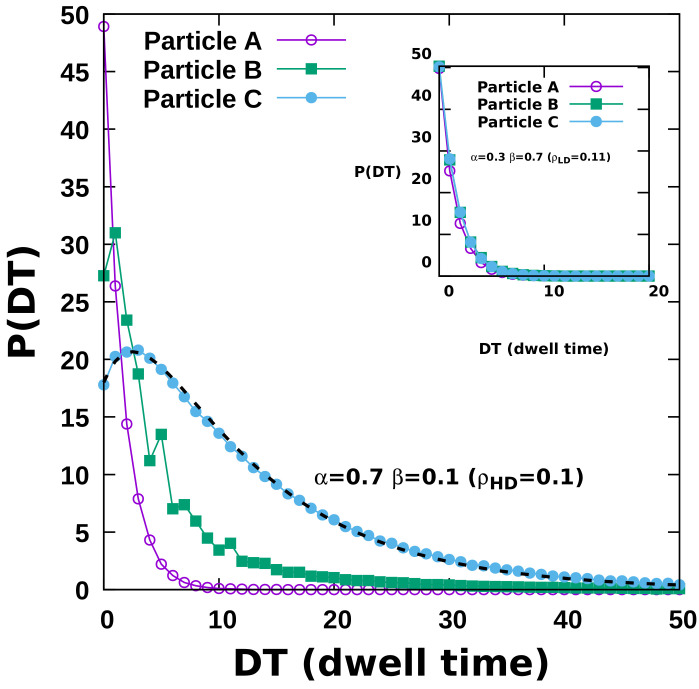
Dwell-time distribution for tagged dynein particles at different entry times in the quasi-two-dimensional system. The main figure corresponds to the HD phase, while the inset displays data for the LD phase. The waiting times for the low-density phase are always governed by a single exponential distribution with a density-dependent decay rate, as shown in the inset. In contrast, the main graph displays prominent double-exponential behavior characteristic of an extremely congested system. The dashed black line shows the best fit to a double exponential, $18(2e^{-0.09x}-e^{-0.32x})$.

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
