# Peer review of "Dynein-Inspired Multilane Exclusion Process with Open Boundary Conditions"

_entropy, 2021, doi:10.3390/e23101343_

Round 1

Reviewer 1 Report

Type of manuscript: Article
Title: Dynein-inspired multilane exclusion process with open boundary
conditions
Journal: Entropy

The Authors study a model for dynein kinetics based on the exclusion process in one and quasi-two dimensions. In one dimension they find a phase diagram that is qualitatively similar to the phase diagram of the standard TASEP, but with differences at the phase boundaries between the low-density and high-density phases and also the size of the maximum-current phase is found to be smaller, which is expected. However, in quasi-two dimensions they find an absence of the maximum-current phase under low loading rate of the ATP and slow longitudinal hopping. This result is new and I find it unexpected and interesting. The paper is well-written and clear, although I have some criticism listed below. Overall I recommend this paper for publication in Entropy, provided the Authors address raised issues. 

Comments:
---------

1. I don't understand how Eq. 5 is derived. If a particle makes i>1 steps in one go, then shouldn't the rate be dependent on observing a gap of size i in front of the particle? In that case the rates should be dependent on (1-rho)^i, rather than on rho. Could you please explain this part better and how these rates had been derived?

2. Although the model is explained very well, especially with the help of Fig. 1, one thing that I find unclear is what happens close to the boundaries? For example, what happens if a particle is at site L_x-2, and it is selected to move 4 sites? Is it allowed to move the selected distance of 4 and be removed from the lattice, or is this move rejected and only a move of size less than or equal to 3 must be selected? 

This is an important question and I strongly believe that the oscillations the Authors observe close to the boundaries are due to this issue. In particular I suspect the oscillations would be absent if the particle is removed in the case the selected distance is equal to or larger than the distance from the boundary. This would be worth checking in simulations.

3. For completeness, it would be worth mentioning how to find rho_max (by maximising J) and writing an implicit equation for rho_max (which if I understand correctly must then be solved numerically due to the strong nonlinearity of J(rho)).

4. The lack of maximum current phase under low and intermediate loading is intriguing. In the paper that is explained by infrequent lane changing. I wonder if the maximum current phase can be restored by increasing the longitudinal hopping rate p_y? This has not been checked in the paper, and may be worth considering, especially if there is no strong biological evidence that p_y << p_x.

5. The statistics of a tagged particle in the TASEP has been considered under various settings: short-time vs long-time behaviour, finite vs infinite system, and fixed vs ensemble averaged initial conditions. The precise law for the tagged particle's position subtly depends on these settings, as discussed in Ref 23. It is not entirely clear what are the initial conditions here, especially for particles B and C. 

6. I do not understand the following paragraph on page 8: "For the dynein particles in our model, the simulation data yield that the MSD obeys a power-law growth with logarithmic corrections in the form t^xi/ln t, where xi takes values in the range between 3/2 to 1 depending on the crowding conditions over the lattice. Irrespective of the load conditions, the dynamics for a single dynein particle is superdiffusive, and the MSD along the drive direction grows algebraically with time with an exponent 3/2. However, in a highly crowded environment, the particle kinetics slows down, and xi approximately takes a value close to unity as shown in Fig. 6." 

Where do we see the xi=3/2 scaling in Figure 6? Also isn't the second sentence in contradiction to the third one? Speaking of Fig. 6, it is not explained what the insets are. 

Typos:
------

1. "explcitly" --> "explicitly" on page 2
2. "alebraically" --> "algebraically" on page 8

Reviewer 2 Report

See attached file
